


# Aerodynamic size-resolved composition and cloud condensation nuclei properties of aerosols in Beijing suburban region

Chenjie Yu[1,2], Dantong Liu[1], Kang Hu[1], Ping Tian[3], Yangzhou Wu[1], Delong Zhao[3], Huihui Wu[2], Dawei Hu[2], Wenbo Guo[2], Qiang Li[4], Mengyu Huang[3], Deping Ding[3] and James D. Allan[2,5]

[1] Department of Atmospheric Sciences, School of Earth Sciences, Zhejiang University, Zhejiang 310027, China

[2] Department of Earth and Environmental Sciences, University of Manchester, Manchester M13 9PL, United Kingdom

[3] Beijing Weather Modification Office, Beijing 100089, China

[4] Cambustion Ltd China Office, Shanghai 201112, China

[5] National Centre for Atmospheric Sciences, University of Manchester, Manchester M13 9PL, United Kingdom

*Correspondence to: Dantong Liu (dantongliu@zju.edu.cn) and James Allan (james.allan@manchester.ac.uk)*



**Abstract** The size-resolved physiochemical properties of aerosols determine their
atmospheric lifetime, cloud interactions, and the deposition rate on human respiratory
system, however most atmospheric composition studies tend to evaluate these
properties in bulk. This study investigated size-resolved constituents of aerosols on
mass and number basis, and their droplet activation properties, by coupling a suite of
online measurements with an aerosol aerodynamic classifier (AAC) based on
aerodynamic diameter ($D_a$) in Pinggu, a suburb of Beijing. While organic matter
accounted for a large fraction of mass, a higher contribution of particulate nitrate at
larger sizes ($D_a > 300$ nm) was found under polluted cases. By applying the mixing
state of refractory black carbon containing particles (rBCc) and composition-dependent
densities, aerosols including rBCc were confirmed nearly spherical at $D_a > 300$ nm.
Importantly, the number fraction of rBCc was found to increase with $D_a$ at all pollution
levels. The number fraction of rBC is found to increase from ~3% at ~90 nm to ~15%
at ~1000 nm, and this increasing rBC number fraction may be caused by the coagulation
during atmospheric aging. The droplet activation diameter at a water supersaturation of
0.2% was $112 \pm 6$ nm and $193 \pm 41$ nm for all particles with $D_a$ smaller than 1 μm
(PM$_1$) and rBCc respectively. As high as $52 \pm 6$% of rBCc and $50 \pm 4$% of all PM$_1$
particles in number could be activated under heavy pollution due to enlarged particle
size, which could be predicted by applying the volume-mixing of substance
hygroscopicity within rBCc. As rBCc contributes to the quantity of aerosols at larger
particle size, these thickly coated rBC may contribute to the radiation absorption
significantly or act as an important source of cloud condensation nuclei (CCN). This
size regime may also exert important health effects due to their higher deposition rate.

## 1. Introduction

Atmospheric aerosols make a significant contribution in a number of atmospheric
chemical and physical processes (Riemer et al., 2019). Aerosols from anthropogenic
emissions have negative impact on air quality and human health (West et al., 2016). As
a major megacity, the air pollution in Beijing and its surrounding regions has raised
much attention in the past years (Shi et al., 2019). The rapid urbanization and the
continued increase in vehicle numbers have contributed to a complicated air pollution



situation in Beijing (Squires et al., 2020). A number of in-situ measurements have
characterised the submicron aerosol compositions in urban Beijing (Wang et al.,
2020;Hu et al., 2016;Wang et al., 2019). However, few studies have detailed
characterised the fine particle compositions or cloud condensation nuclei (CCN)
abilities at Beijing rural sites (Chen et al., 2020a). The relocation of industry from the
urban Beijing has led the surrounding cities around Beijing to be highly industrialised
in recent years (Wang et al., 2018), and the rural sites of Beijing are significantly
impacted by the air pollutants transported from the surrounding industrial regions in
North China Plain (NCP) (Wu et al., 2011). Furthermore, controls targeting pollution
from residential solid fuel use and diesel vehicles do not apply outside of the main
metropolitan area of Beijing. Detailed characterisation of fine aerosol physiochemical
properties in a variety of different environments is essential to understand the evolution
of atmospheric particulate matter.

Fine particulate matter can also cause damage to human respiratory system (Xing et al.,
2016;Xu et al., 2016). Aerodynamic size of aerosols crucially determines their
deposition to the respiratory system (Sturm, 2010;Vu et al., 2018;Sturm, 2017), e.g.
particle with aerodynamic diameter ($D_a$) below 2.5 μm can reach small air ways of
respiratory system and further pass into the blood (Lipworth et al., 2014). The observed
toxicity of aerosols is composition dependent (Kwon et al., 2020) and influenced by the
complex morphology (Sturm, 2010;Vu et al., 2018), therefore the aerodynamic size-
resolved properties of aerosols are important to understand their influences on air
quality associated with the human health.

Atmospheric aerosols also play important roles in climate through scattering and
absorbing solar radiation directly, or indirectly through altering cloud properties (Liu
et al., 2020;Ravishankara et al., 2015). Black carbon (BC) is produced from incomplete
combustion and is the dominant optically absorbing component in aerosols (Liu et al.,
2020;Bond et al., 2013). By mixing with other compounds, the absorption ability of
coated BC can be enhanced through the "lensing effect" (Lack and Cappa, 2010).
However, detailed simulation and characterization of optical properties of BC remains



large uncertain since it is influenced by factors such as shape and mixing state (Cappa
et al., 2012;Liu et al., 2017;Fierce et al., 2020), which can be modified through
atmospheric processing. Thus, better characterization of light absorbing carbonaceous
particles is essential.

Coated BC is also known as an important source of CCN and wet removal is its main
atmospheric loss mechanism, so its in-cloud scavenging efficiency and thus lifetime is
influenced by its size, mixing state and hygroscopic properties (Taylor et al., 2014), but
this is subject to large uncertainties (Myhre and Samset, 2015). Studies have confirmed
that hygroscopicity of rBCc is largely impacted by the coating material, and rBCc will
transform from hydrophobic to hydrophilic after emission by acquiring more non-BC
material and increasing in size (Hu et al., 2020b;Wu et al., 2019;Liu et al., 2013).
However, limited studies (Levin et al., 2014;Broekhuizen et al., 2006) have provided
both the measured size-resolved CCN ability and aerosol physiochemical properties so
far, and the CCN ability of rBCc based on atmospheric data remains poorly constrained.
Previously, size-resolved composition has been widely investigated using size-
segregated off-line analysis of cascade impactors samples (Marple et al., 1991). This
technique offers great advantages in obtaining detailed information about compositions
in combination of the advanced offline measurements, however it often requires large
amounts of material and may not be able to provide sufficient time resolutions.
Information about particle mixing state and CCN activities are also not available
through this technique. The aerosol mass spectrometer (AMS) is also capable of
delivering size-resolved aerosol compositions, however the poor accuracy of AMS in
the size range important for CCN (typically 50 -100 nm) has hampered quantitative
work for this application.

A previous study (Yu et al., 2020) has characterized the size-resolved rBCc mixing
state in Beijing using a tandem aerosol classifier system. To explore the size-resolved
physiochemical properties and CCN ability for bulk aerosol compositions, here we
performed a new online measurement method by coupling an aerodynamic aerosol




classifier (AAC) with different aerosol measurement techniques including a single-
particle soot photometer (SP2) and an aerosol mass spectrometer (AMS). Comparing
to the previous studies performed with differential mobility analyzer (DMA), the AAC
classifies particles without multiple charging artefacts in a wide size range and with
better transmission efficiency (Johnson et al., 2018). The simultaneous measurement of
size-resolved chemical composition and CCN activation enables a detailed analysis of
rBCc hygroscopicity and its size-dependent contribution to the CCN activation in a
polluted environment. This information will deliver a better understanding to the BC
deposition properties for the climate and air pollution impact on human health studies.

## 2.  Experimental methods
### 2.1 Experiment location and instruments
The experiment was performed between 5th Jan and 20th Jan 2020 in the Beijing
Weather Modification Office field experiment base located in Pinggu (the red
pentagram shown in Fig. 4a), a northeastern suburb of Beijing (Shi et al., 2019). With
agriculture dominated its local economy, Pinggu is surrounded by small villages and
farmlands (Han et al., 2020). Fig. 1 describes the schematic of the instruments used for
the size-resolved aerosol measurements. An aerodynamic aerosol classifier (AAC,
Cambustion) was placed upstream of the aerosol measurement instruments. The
operation and validation of the AAC was described in previous studies (Tavakoli and
Olfert, 2013;Tavakoli and Olfert, 2014). Unlike the DMA or Centrifugal Particle Mass
Analyzer (CPMA), the AAC selects particles based on aerodynamic sizes according to
particle relaxation time without needing charging for electrostatic or mass sizing. A
suite of online measurements was introduced downstream of the AAC, including a
high-resolution time of flight aerosol mass spectrometer (HR-ToF-AMS, Aerodyne)
(DeCarlo et al., 2006) was operated in V-mode to characterize the non-refractory
aerosol composition and a single particle soot photometer (SP2, DMT) (Schwarz et al.,
2010) for the measurement of rBCc concentrations. The volume properties of non-
refractory material within rBCc (hence referred as 'coating thickness') was derived by
SP2 leading-edge-only (LEO) method (Liu et al., 2019) and is described as the ratio
between the diameter of total rBCc and rBC core ($D_p/D_c$). A cloud condensation nuclei


counter (CCNc, DMT) was used to sample the potential CCN activation ability at a
constant supersaturation (SS) of 0.2% and a condensation particle counter (CPC, TSI
model 3776) was used to measure the condensation nuclei (CN) number concentration.
The SP2 incandescence signal was calibrated using nebulised Aquadag black carbon
particle standards while the scattering channel was calibrated by 200 nm polystyrene
latex spheres before the measurement, and the correction factor of 0.75 for ambient rBC
measurement was applied (Laborde et al., 2012). The ionization efficiency of AMS was
calibrated using mono-disperse of nebulized ammonium nitrate particles following the
standard protocols (Xu et al., 2017), and a constant collection efficiency (CE) of 0.5 is
applied (Middlebrook et al., 2012). More details of the calibration and operation of this
AMS instrument can be seen in previous field measurement studies (Hu et al., 2021;Liu
et al., 2021).  The term 'all particles' in this study is referred as the $PM_1$ compositions
including Organic compounds (Org), Sulfate ($SO_4$), ammonia ($NH_4$), Nitrate ($NO_3$),
Chloride (Cl) and rBC from AMS and SP2. The AAC was set to classify dry aerosol
particles from 90 nm to 1100 nm in aerodynamic diameter ($D_a$) to cover the detection
range of the SP2 and AMS. It took around 15 min to complete one scan using the AAC
step scanning mode, and a timed valve was placed at the upstream of the AAC for
switching between monodisperse and polydisperse every 30 min. Example for a
running cycle is presented in the supplementary.

## 2.2 Calculation of aerosol morphology parameters

The dynamic shape factor ($\chi$)  describes the shape of particles (DeCarlo et al., 2004).
$\chi = 1$ denotes a perfectly spherical particle, and $\chi > 1$ means more non-sphericity.
Based on the measurement here, $\chi$ can be calculated by:
$$\chi = \frac{\rho_p D_v{}^2 C_c(D_v)}{D_a{}^2 C_c(D_a)} \tag{1}$$

where $\rho_p$ is the particle material density, $C_c$ represents the slip correction factor at a
given diameter and is calculated following the description in Kim et al. (2005), $D_v$ is
the particle volume equivalent diameter, and $D_a$ is the aerodynamic diameter classified
by AAC. This calculation is performed for all particles (including rBCc) and rBCc,
using their respective parameters ($\rho_p$ and $D_v$). For all particles, $\rho_p$ is the mean density
weighted by the $PM_1$ results measured by the AMS and SP2. To compute the particle
volume results based on the AMS measured ion and Org concentrations, a simplified



ion pairing scheme presented in Gysel et al. (2007) was applied, and the solutions are
described in the supplementary. The $\rho_p$ of rBCc is calculated as the weighted density
within rBCc including rBC and coatings, where the coating material of rBCc is assumed
to constitute of the same volume fractions of ambient non-refractory compositions (Liu
et al., 2015;Hu et al., 2021):
$$\rho_{\text{rBCc}} = \frac{M_{\text{rBCc}}}{V_{\text{rBCc}}} = \frac{\rho_{\text{NR}} \cdot \left(\frac{1}{6}\pi D_{\text{p,rBCc}}^3 - \frac{1}{6}\pi D_c^3\right) + M_{\text{rBC}}}{\frac{1}{6}\pi D_{\text{p,rBCc}}^3} \tag{2}$$

where $M_{\text{rBCc}}$ and $V_{\text{rBCc}}$ are the mass and volume of the rBCc respectively, $\rho_{\text{NR}}$ is the
particle density of non-refractory compositions. The rBC core diameter ($D_c$) and total
rBCc diameter ($D_{\text{p,rBCc}}$) are derived through the SP2 LEO method.
For all particles, mean single particle mass is derived from the total mass ($M_{\text{all}}$)
obtained by AMS and SP2 divided by the total number ($N_{\text{total}}$) obtained by the CPC,
hereinafter the mean $D_v$ of particle is assumed to equal to the mass equivalent diameter
($D_m$) and is obtained by applying the mean $\rho_p$ above:
$$D_{v,\text{all}} = D_{m,\text{all}} = \sqrt[3]{\frac{6M_{\text{single,all}}}{\rho_{\text{all}} \cdot \pi}} = \sqrt[3]{\frac{6M_{\text{all}}}{\rho_{\text{all}} \cdot \pi \cdot N_{\text{total}}}} \tag{3}$$

where $\rho_{\text{all}}$ is the particle density of all aerosol particles.

**2.3 Hygroscopicity parameter calculation**
The hygroscopicity parameter ($\kappa$) (Petters and Kreidenweis, 2007) of measured
aerosols is predicted based on the measured aerosol compositions and invoking the
Zdanovskii–Stokes–Robinson (ZSR) mixing rule(Stokes and Robinson, 1966). The $\kappa$
for all particles ($\kappa_{\text{all}}$) is calculated as:
$$\kappa_{all} = \sum_i \varepsilon_i \kappa_i \tag{4}$$

where $\varepsilon_i$ and $\kappa_i$ is the volume fraction and hygroscopicity parameter of each chemical
composition respectively. The $\kappa$ based on the AMS measured concentrations were
calculated based on the same simplified ion pairing scheme described above. The
detailed information for each parameter used for $\kappa$ calculation is listed in Table S1. For
rBCc, the $\kappa_{\text{rBCc}}$ is calculated by:
$$\kappa_{rBCc} = \sum_i \varepsilon_{coating,i} \kappa_{coating,i} + \varepsilon_{rBC} \kappa_{rBC} \tag{5}$$





Where $\varepsilon_{\text{coating},i}$ and $\varepsilon_{\text{rBC}}$ is the volume fraction coating and rBC respectively; $\kappa_{\text{coating},i}$
represents the hygroscopicity parameter for each rBCc coating composition and is
assumed to equal to the $\kappa$ of ambient non-refractory compositions (Motos et al.,
2019;Hu et al., 2021); $\kappa_{\text{rBC}}$ represents the hygroscopicity parameter for rBC and the
last term can be ignored since pure rBC is assumed to be hydrophobic.

**2.4 CCN ability of all particles and rBCc**

The CCN activation fraction is determined as the ratio between CCN number
concentration at SS=0.2% and the CN number concentration measured by the CPC. The
size-resolved CCN activation fraction ($AF$) is fitted in a sigmoid function:
$$AF = \frac{E}{1+\left(\frac{D_{50}}{D_p}\right)^C} \times 100\% \qquad (6)$$

Where $E$ and $C$ are fitting coefficients which represent the asymptote and the slope
respectively. $D_p$ is the particle dry diameter, and $D_{50}$ represents the critical particle
diameter where 50% of particles in number can be activated as CCN (Petters and
Kreidenweis, 2007).

The number concentration of rBCc which acts as CCN is derived from the concurrent
measurements of rBC number concentration, CCN and CN. The method described by
Hu et al. (2021) has been applied to determine the activation of rBCc. Firstly, the un-
activated particle number concentration is derived from the difference between CN and
CCN, as the red line in Fig. 2(a) shows.  For particles with $D_a > 300$nm in the example,
the un-activated particles are nil thus all rBCc is also activated. Here particles are
considered to be well mixed, and rBCc is less hydrophilic than any other non-refractory
particles at the same particle size. Thus, the rBCc is more difficult to be activated as
CCN than the other particles. For particles with $D_a < 300$nm, the rBCc is therefore
considered to be the first in contributing the un-activated particles and the activated
rBCc is the rBCc number concentration higher than the un-activated particle numbers.
By this way, the size-dependent activated rBCc number concentration can be obtained
(black line in Fig. 2(b)). $D_{50,\,\text{rBCc}}$ can then be derived through Equation (6) based on the
rBCc activation fraction curve. The rBCc activation fraction derived through this
method is further referred as "measured $AF_{\text{rBCc}}$". There may be some occasions when
rBCc could exhibit a higher hygroscopicity, if coated with sufficient hygroscopic


substances, even higher than a particle without containing rBC. This means the scenario
here may underestimate some fractions of activated rBCc. The method here may
therefore serve as a least estimation of droplet activation of rBCc from this aspect.

The rBCc activation is also estimated through the calculated size-resolved critical
supersaturation ($SS_c$) (Wu et al., 2019;Hu et al., 2021) for comparsion, which is derived
based on the $\kappa_{rBCc}$ described before from the $\kappa$-Köhler theory:
$$S(D) = \frac{D^3 - D^3_{rBCc}}{D^3 - D^3_{rBCc}(1-\kappa)} exp\left(\frac{4\sigma_{s/a}M_w}{RT\rho_w D}\right) \tag{7}$$

Where $D$ is the diameter of the droplet, $D_{rBCc}$ is the rBCc dry diameter, $M_w$ and $\rho_w$ are
the molecular weight and density of water respectively, $T$ is temperature, $R$ is the idea
gas constant, and $\sigma_{s/a}$ is the surface tension of the solution/air interface. A decreased
SS with increasing $D_a$ can be obtained (Fig. 3), so the $D_{50, rBCc}$ at $SS_c = 0.2\%$ is the cross
point above which diameter only SS<0.2% is required to activate the targeting rBCc.
The activated rBCc number concentration is the rBCc concentration with size larger
than $D_{50, rBCc}$. The activation fraction estimated through this method is further referred
as "modelled $AF_{rBCc}$".

**2.5 NAME dispersion model**
The airmass classification results used to identify potential source regions are generated
by the UK Met Office Numerical Atmospheric dispersion Modelling Environment
(NAME) dispersion model (Jones et al., 2007). The model presented the 48h backward
dispersion results on a $0.25° \times 0.25°$ grid using the three-dimensional gridded
meteorological field generated from the UK Met Office's Unified Model (Brown et al.,
2012). Beijing and its surrounding areas have been classified into five regions as shown
in Fig. 4(a) in order to attribute the airmass histories: The Local Beijing (39-41.5°N,
115-117°E), the North (41.5-45°N, 104-121°E), the South (32-39°N, 115-121°E), the
West (32-41.5°N, 104-115°E) and the East region (39-41.5°N, 117-121°E).

**3.   Results and discussions**
**3.1 Overview for the whole campaign period**
Fig. 4(c) presents the overview of aerosol total number and mass concentrations during
the experimental period. Beijing and its suburban region experience large contrasts in



pollution conditions depending on the wind direction (Liu et al., 2019;Chen et al.,
2020b). To test whether the mixing state varies according to ambient pollution
concentrations, the pollution is classified into three levels according to the frequency
distribution of $PM_1$ concentrations during the whole measurement period: heavy
pollution ($PM_1 \geq 30$ µg/m$^3$), moderate pollution ($10$ µg/m$^3 < PM_1 < 30$ µg/m$^3$), and
light pollution ($PM_1 \leq 10$ µg/m$^3$). Combining the airmass history results with the
aerosol optical depth (AOD) spatial distribution results from the Himawari-8 Level 2
aerosol product (Bessho et al., 2016;Fukuda et al., 2013) (Fig. 4(b)), the heavy and
moderate pollution period was mostly attributed to airmasses from the East and West
regions. While the contribution from the Local airmass cannot be ignored in some
pollution cases, relatively clean northerly airmasses were associated with the light
pollution periods.

## 3.2 Size-resolved aerosol mass compositions and rBCc mixing state

Fig. 5(a-e) presents the size-resolved average mass concentrations for $PM_1$, rBC,
organic compounds (Org), sulfate ($SO_4$), nitrate ($NO_3$) respectively under each
pollution condition. Though the heavy pollution period has the highest aerosol mass
concentrations among three cases, there was no significant difference for total $PM_1$ and
non-refractory compositions mass concentrations below 200 nm in $D_a$ under different
pollution levels. Notable contributions to the total $PM_1$ from non-refractory material
was observed for $D_a > 300$ nm especially for the heavy pollution condition. Unlike the
more polluted conditions, the non-refractory aerosol mass concentrations during light
pollution periods shows limited size-dependent variation. The size distribution of rBC
mass concentration reached the peak at 400 nm under heavy pollution, while the peak
for the moderate and light pollution was at a bit smaller $D_a$ which was between 300 and
400 nm. The peak diameter of NR-$PM_1$ observed in Pinggu was at around 700 nm and
is higher than the peak diameter of NR-$PM_1$ reported at the urban site of Beijing which
is between 400 nm and 500 nm in winter (Hu et al., 2016). Due to the higher primary
organic aerosol (POA) emissions, the results at Beijing urban site has higher
contribution of Org at smaller size (< 500 nm) (Zhang et al., 2014). While the higher
Org peak diameter (at around 700 nm) shown in our study suggests that the Org was
highly oxidised in Beijing suburban (Li et al., 2021). The higher peak diameter of
secondary inorganic compound also indicates the well mixture of aerosol components



in suburban region (Liu et al., 2016). This size-resolved composition result reported in
Pinggu is in consistent with the previous measurement in another suburban region in
NCP (Li et al., 2021). Comparing the composition mass fractions under three different
pollution cases shown in Fig.5(i-k), one of the remarkable differences is that particulate
nitrate accounted more mass fraction during the heavy and moderate pollution periods
than during the light pollution period. Previous studies shown that this rapid particulate
nitrate formation in Beijing area is mainly associated with the heterogeneous hydrolysis
of $N_2O_5$ at night (Li et al., 2018). Particulate nitrate has become one of the major
secondary inorganic aerosol pollutants in urban environment recently (Zhang et al.,
2015), and $NO_3$ also contributed to the aerosol hygroscopicity significantly during the
haze pollution periods (Sun et al., 2018). Due to the significant reduction of $SO_x$
emissions in China in recent years (Zhang et al., 2012), the mass fractions of $SO_4$
remained low in pollution cases. The Org contributed to the aerosol mass compositions
significantly, and the capping of rBC mass fraction was around 25% among all three
cases.

Fig. 5(f) and (g) present the size distribution of rBC core mass median diameter (MMD)
and the coating thickness. This indicates larger $D_a$ had selected rBCc with larger rBC
core and higher coatings. The MMD of rBC core increased from below 100 nm to
around 300 nm with the increasing of particle size. The rBC core for the light pollution
condition was a little smaller than the other two periods, indicating a possible
coagulation process in more polluted cases with higher rBC concentrations. The coating
thickness of rBCc $D_p/D_c$ decreased slightly when $D_a$ increased from 90 to 300 nm. This
decreasing trend of rBC coating thickness may be caused by the traffic emissions. Joshi
et al. (2021) demonstrated that traffic emissions dominated the rBC fluxes in urban
Beijing, and a previous study also showed a similar decreasing trend of rBC coatings
for engine emissions within this particle size range (Han et al., 2019). Limited
differences were observed for the size-resolved $D_p/D_c$ among the three pollution levels.
The average $D_p/D_c$ for all rBCc was 2.1 ± 0.2, 1.6 ± 0.1, and 1.5 ± 0.04 for heavy,
moderate and light pollution respectively. There was more heavily coated rBCc showed
for heavy pollution condition, and this was consistent with more secondary particle
formation than the other periods.



Fig. 5(h) shows the distribution of hygroscopicity parameter ($\kappa$). The lowest $\kappa_{all}$
between 150 and 300 nm at heavy and moderate pollution condition was mainly caused
by the increasing of rBC fractions. Due to the increase of more hydroscopic inorganic
compositions for larger particles under heavy and moderate pollution conditions, $\kappa_{all}$
increased considerably for particles $D_a > 200$ nm and 300 nm. In contrast to the more
polluted cases, $\kappa_{all}$ under light pollution period varied slightly with the increase of $D_a$.
Caused by the absence of more soluble inorganic compositions, $\kappa_{all}$ for particles with
$D_a > 300$ nm during light pollution period was lower than the other conditions. For
rBCc, $\kappa_{rBCc}$ was more influenced by the coating volume fractions rather than the
coating compositions, as the variation of $\kappa_{rBCc}$ generally followed the trend of rBCc
coating thickness (Fig. 5(g)). $\kappa_{rBCc}$ for particles with $D_a < 300$ nm was close under three
different pollution levels, and the decreasing trend of $\kappa_{rBCc}$ between 90 and 300 nm
was caused by the reduction of coating material fraction.

**3.3 Size-resolved particle morphology**
Fig. 6 shows the distribution of particle density, average single particle size and mass,
and morphology parameters for all particles (left) and rBCc (right). The average particle
density for all particles ($\rho_{all}$) varied slightly between 1.55 and 1.6 g/cm$^3$, and the rBCc
particle density ($\rho_{rBCc}$) within the measurement size range was generally higher than
the $\rho_{all}$ due to the higher density of rBC. The peak $\rho_{rBCc}$ reached at between 200 and
300 nm in $D_a$ due to the rBCc was least coated within this size range. $D_v$ was larger
than $D_a$ and deviated more at smaller size but was close to $D_a$ for all particles and rBCc
larger than 200 nm. The dynamic shape factor ($\chi$) of all particles declined from around
1.8 to 1.2, while $\chi$ of rBCc declined from around 2 to 1.2. All particles with $D_a$ above
400 nm and rBCc with $D_a$ above 500 nm tended to have lower $\chi$ which was around 1.2.
Previous study (Lin et al., 2015) in other megacities reported that $\chi$ of all particles was
around 2 with $D_a$ at around 100 nm which is close to our results. Previous measurements
(Zhang et al., 2016) in Beijing also shown the similar decreasing trend of $\chi$ for rBC
core during the aging process.

This result indicates that smaller particles have more irregular shapes, while particles
with larger aerodynamic size are more spherical in ambient atmosphere. Previous
experiment shows that the irregular aggregated rBCc from fresh emissions can





transform to be more spherical-like by acquiring more secondary species(Ahern et al.,
2016). Our results confirm that the spherical assumption is suitable for large rBCc in
aerodynamic size in a typical anthropogenic polluted environment. This also implies
that larger and spherical particles tend to have larger deposition rate, while particles
with more irregularity may experience higher drag force in the air, towards decelerating
the settlement.

**3.4 Size-resolved CN and CCN number concentrations**
Fig.7(a) and (b) presents the distribution of rBCc, CN and CCN number concentrations
at different polluted conditions. The peak of rBCc number concentration at heavy
pollution period was at around 300 nm, while the peak for moderate and light pollution
was slightly smaller (at around 200 nm). This agrees with the previous studies in Beijing
showing that the average total rBCc particles size was associated with the pollution
levels (Yu et al., 2020;Liu et al., 2019). Similar trend was also observed for the CN
concentrations, and the peak of CN concentrations shifted to the larger particle size with
increased pollution level. Higher levels of secondary production through condensation
and also coagulation enlarged particle size.   Because of the increasing of average
particle size, more fraction of particles can be activated as CCN under heavy pollution
period.

By using aerodynamic size-resolved number concentration of rBC and CN, a
remarkable increase of rBC number fraction at larger aerodynamic size was found (Fig
7(c)), i.e. with $D_a$ from 100-1000nm, rBC number fraction increased almost linearly
from 3% to 15%, and this applied to all pollution levels. This tends to represent a
generic phenomenon for a suburban environment with continuous influence of
anthropogenic emissions, and the primary emissions had been aged in a time scale of
hours. Fine rBC condensed on or coagulated with pre-existing larger particles during
the aging process (Riemer et al., 2009). The coagulation process dominated the
formation of thickly-coated rBC particles (Reddington et al., 2013), and the coagulation
rate of smaller rBC may be fast due to the higher number concentration of fine mode
particles (Matsui et al., 2018). The very fresh emissions such as from diesel engine
emissions, which mostly contains thinly-coated small rBC (Han et al., 2019), may not
show the same rBC number fraction distribution. Previous studies also reported a



relatively fast aging process for BC also in the order of hours (Peng et al., 2017), if
under a polluted environment rich of precursors, the aging could be even faster (Peng
et al., 2016). The causality of this increased rBC number fraction at larger particle size
is therefore the non-rBC compounds associated with it. The results presented here
indicated that the higher contribution from regional pollution to the rBC number at
larger aerodynamic size may apply, albeit the various features of primary sources in
winter (Wang et al., 2019;Liu et al., 2019).

The rBC associated with larger coatings was more spherical (with $\chi$ close to 1, as
discussed above), therefore more likely to have a core-shell structure, which would lead
to an absorption enhancement from the lensing effect of coatings (Liu et al., 2017). In
addition, this size-resolved rBC number fraction results will improve the understanding
to the lung deposition of BC in human health studies (Rissler et al., 2017). This means
for particles with higher deposition rate tend to contain a higher number fraction of
rBC, which may provide some indications for constituents deposited on different parts
in human respiratory system (Carvalho et al., 2011;Manigrasso et al., 2020).


### 3.5 CCN ability and rBCc activation

Presented in Fig. 8(a), the $D_{50,\,\text{All}}$ varied smoothly and was slightly higher than 100 nm
for most of the experiment period. The mean $D_{50,\,\text{All}}$ and $D_{50,\,\text{rBCc}}$ was $112 \pm 6$ nm and
$193 \pm 41$ nm respectively. Shown in Fig. 8(d) most of the $D_{50,\,\text{rBCc}}$ was around 200 nm
which illustrates that the number concentrations of rBCc with $D_a$ above 200 nm had
significant contribution to the overall $AF_{\text{rBCc}}$.

Fig. 8(b) presents the temporal evolution of the CCN number concentration and
activation fraction for all particles and rBCc. Fig. 8(e) showed $50 \pm 4\%$ of the measured
particles can be activated with SS=0.2% under heavy polluted period, while the $AF_{\text{all}}$
for the light pollution period was generally lower than the $AF_{\text{all}}$ of the other two periods
which was $24 \pm 10\%$ on average. The $AF_{\text{all}}$ for the moderate pollution period was $39 \pm$
$9\%$ on average. Shown in Fig. 8(e) and (f), both all particles and rBCc showed high
activation fraction of around 50% during the heavy pollution periods. While for
moderate and light pollution conditions, rBCc exhibited substantially higher activation


fraction than all particles, especially under light pollution periods where the average
activation fraction was $44 \pm 18\%$ for rBCc compared to $24 \pm 10\%$ for all particles. The
maintaining high rBCc activation fraction at all pollution levels resulted from the
relatively higher rBC number fractions at larger $D_a$ (Fig. 8(c)) because of the higher
associated coatings. The directly measured CCN activity of rBCc showed that particles
at larger sizes had contained a larger fraction of rBCc that were CCN active, due to the
larger particle size. This in turn implies that the rBCc has the potential to be more
efficiently incorporated into cloud droplets. The measured and modelled $AF_{rBCc}$ was
close and agreed within 22% (shown in Fig. S3), and the modelled $AF_{rBCc}$ was slightly
higher than the measurement results. This underestimation of modelled $D_{50, rBCc}$ may
result from an overestimation on the $\kappa_{rBCc}$ as here a consistent $\kappa$ was applied between
rBCc coatings and all non-refractory materials in bulk, though the coatings on rBC may
have not contained as much hygroscopic materials as the bulk non-rBC aerosols.
Freshly emitted rBC particles contain substantial amounts of organic matter (Peng et
al., 2017) while the more hygroscopic secondary inorganic materials require
atmospheric aging to be mixed with rBC (Hu et al., 2020a). Our results confirm that
while rBCc can be CCN active, and the size of rBCc is crucial to the rBCc CCN ability
in polluted suburban environment. This $AF_{rBCc}$ result presented here is generally
consistent with previous field measurements: Wu et al. (2019) and Hu et al. (2021)
reported 59% and 60% of total rBCc could be activated with SS = 0.2% respectively in
other anthropogenic polluted environment.

## 4. Atmospheric implications

The AAC combination applied in this study introduced a new way to explore the
physiochemical properties of aerosols. The comprehensive size-resolved aerosol
information presented in this study can contribute to future studies focusing on the BC
evolution and lifetime, and improve the particle resolved model simulations (i.e.
Riemer et al. (2009)) for the anthropogenic polluted environment. Importantly, our
results shown that thickly coated rBCc accounted higher number fraction at larger
particle size than the smaller particle size in Beijing suburban. As indicated in Fig. 9,
the mass absorption coefficient (MAC) of rBCc at 880nm wavelength is calculated
through the core-shell Mie model described in the supplementary. The $MAC_{880}$ was
largely enhanced for rBCc with $D_a > 500$ nm. These larger rBCc with higher absorption


efficiency importantly contributed to the total absorption. When transported into the
top of boundary layer, these highly coated and absorbing rBCc can be efficiently
incorporated into clouds (Ding et al., 2019). The absorption effects of these rBCc will
be further magnified by mixing with the cloud water droplets (Wu et al., 2016), and the
lensing effect may reduce the cloud lifetime (Ramanathan et al., 2001) or the cloud
albedo (Chuang et al., 2002). In addition to the strong radiative absorption, these large
rBCc may also alter the regional precipitation rate (Johnson et al., 2019).

**Conclusions**
In this study, a new aerodynamic size selection technique was applied for the direct
size-resolved characterisation of aerosol constituents and properties on both mass and
number basis in a suburban Beijing in winter. Besides the size selection without relying
on particle charging efficiency, this technique allows reliable size-resolved particle
properties. Organic compound accounted around 40% of the total $PM_1$ mass, and we
found higher contribution of particulate nitrate at larger sizes under polluted cases in
Beijing suburban. In particular, particles with larger aerodynamic diameter ($D_a$) were
found to contain a higher number fraction of refractory black carbon (rBC), which
means rBC could be more efficiently mixed with larger particles during atmospheric
processes. Mie calculation results show that these thickly coated rBC containing
particles (rBCc) as included in large particle may have an up to 2-fold of enhanced
absorption. The dynamic shape factors for both refractory and non-refractory particles
have also been derived. Particles with $D_a$ larger than 300 nm tended to have a more
spherical-like shape, while smaller particles were with more irregular shape in the
polluted environment. By applying the method introduced by Hu et al. (2021), as high
as 46 ± 15% number fraction of rBCc was observed to be activated under SS=0.2%.
Our results suggest that the size of rBCc is key to the cloud condensation nuclei (CCN)
activities of rBCc. Though rBC was small and hydrophobic initially, after being mixed
with non-refractory compositions and becoming larger, the rBCc can become CCN
active. The higher number fraction of rBCc at larger particle size ($D_a$ > 300 nm)
emphasizes the importance of the rBCc as a considerable CCN source. In summary, the
rBCc from anthropogenic emissions, after short aging in regional scale, may therefore
alter the regional radiative forcing directly or indirectly through altering cloud
properties and deposit on human respiratory system efficiently.



## Acknowledgments

This research was supported by the National Key Research and Development Program of China (2016YFA0602001, 2019YFC0214703) and the National Natural Science Foundation of China (41875167). The authors acknowledge the Cambustion Ltd for providing the AAC instrument.

## Author Contribution

CY and DL deigned the experiments and wrote the paper; DL and JDA provided guidance with the analysis and writing; CY, DL, KH, PT, YW, DZ, WG, MH and DD performed experiments; CY, DL, KH, HW, DH and JDA contributed to the data analysis; QL provided the AAC and guided the operations.

## Data availability

Raw data is archived at Zhejiang University and is available on request.

## Competing financial interests

The authors declare no competing financial interests.

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






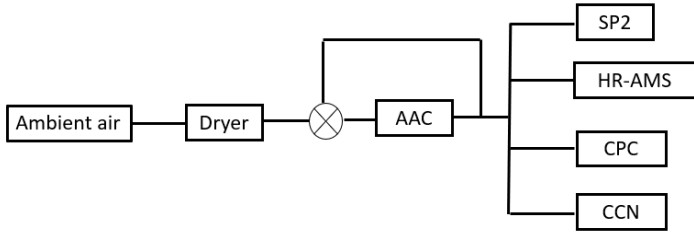


Figure 1. The schematic of the instruments set up. A timed three-way valve was
placed at the upstream of the AAC.

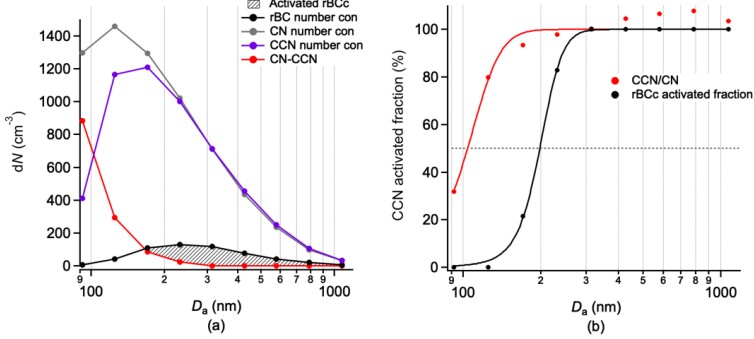


Figure 2 An example of all particles and rBCc activation, the dashed grey line in (b)
indicated the 50% of all particles or rBCc activated.

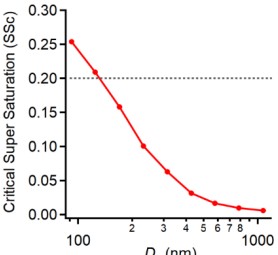


Figure 3 An example of the calculation of the size-resolved critical supersaturation
(SSc)

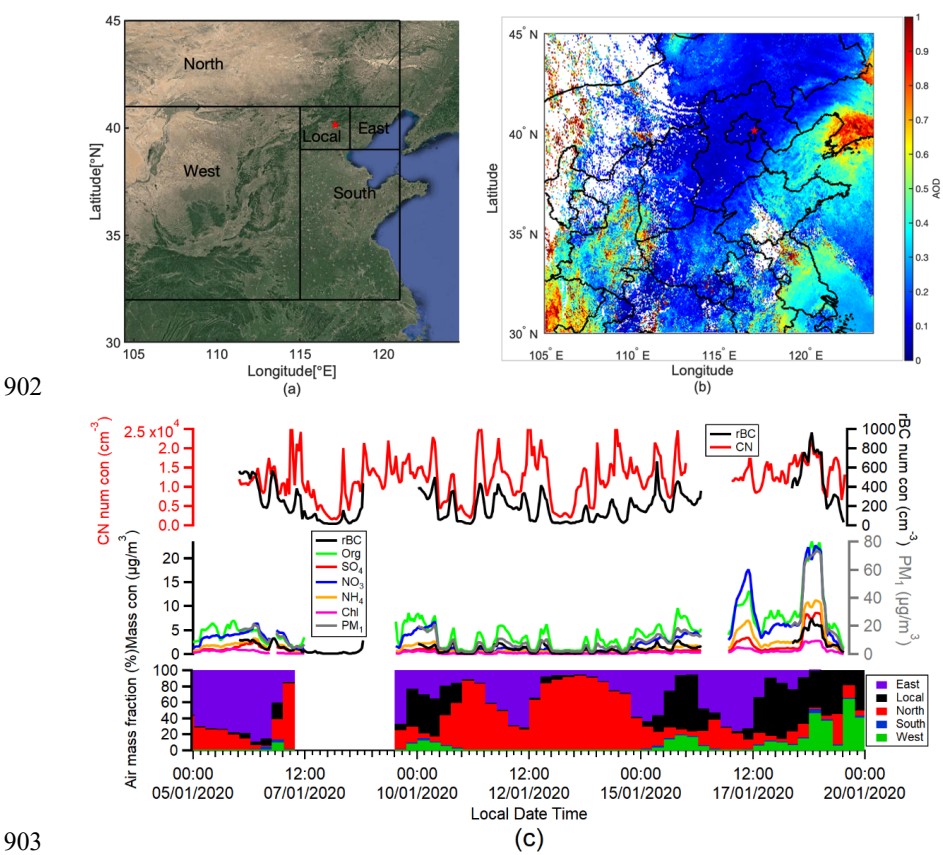



Figure 4. Overview of the experiment. (a) Location of the measurement site (marked in red pentagram) and regions classified for air mass; (b) The mean aerosol optical depth (AOD) distribution during the experiment period; (c) Aerosol mass and number concentrations and classified air mass types.



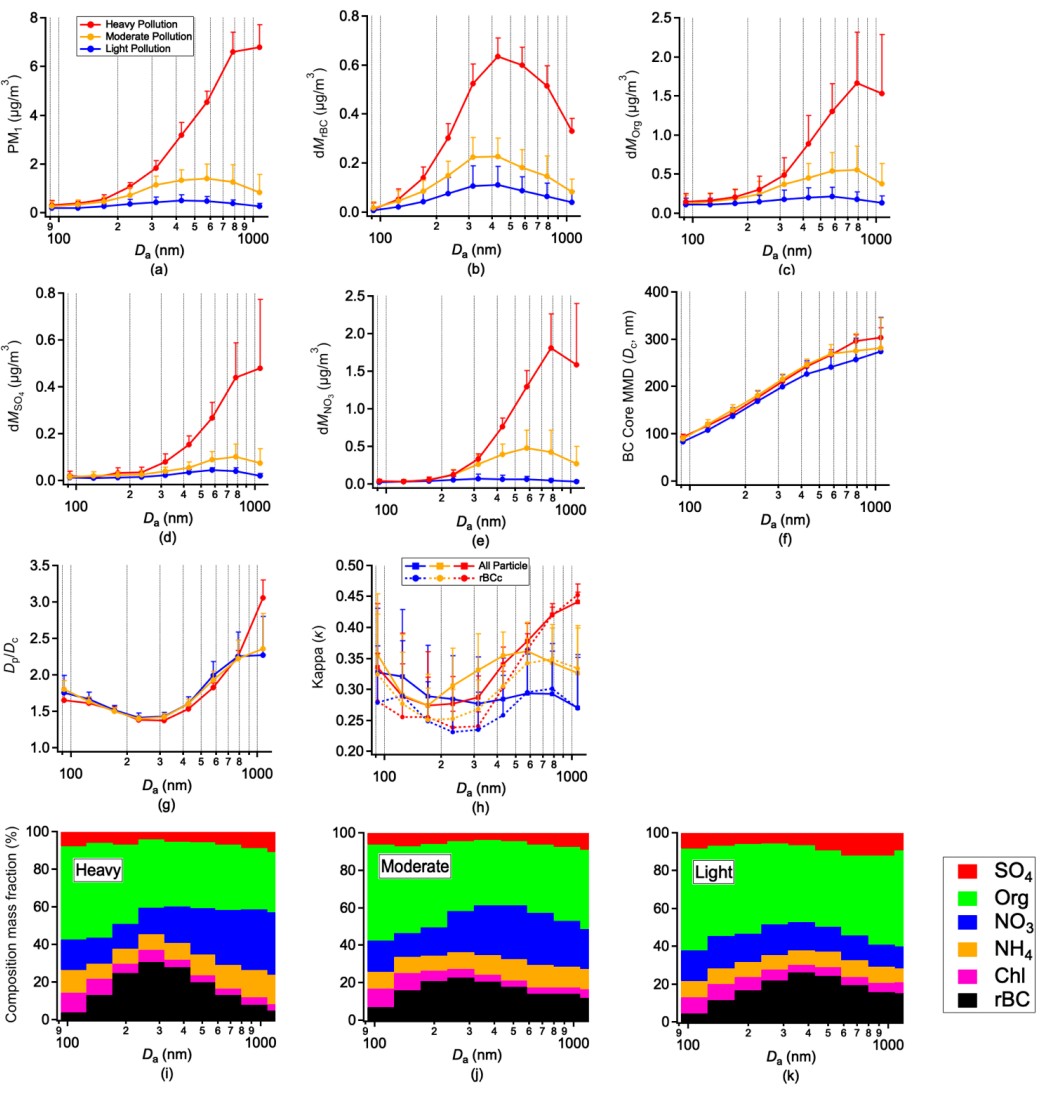

Figure 5 Size-resolved (a) $PM_1$ (mean $\pm$ standard deviation); (b) rBC mass concentration; (c) Org mass concentration; (d) $SO_4$ mass concentration; (e) $NO_3$ mass concentration; (f) size-resolved rBC core mass median diameter (MMD); (g) size-resolved coating thickness ($D_p/D_c$) of rBCc; (h) hygroscopicity parameter ($\kappa$); (i, j, k) aerosol composition mass fractions under three different pollution levels.

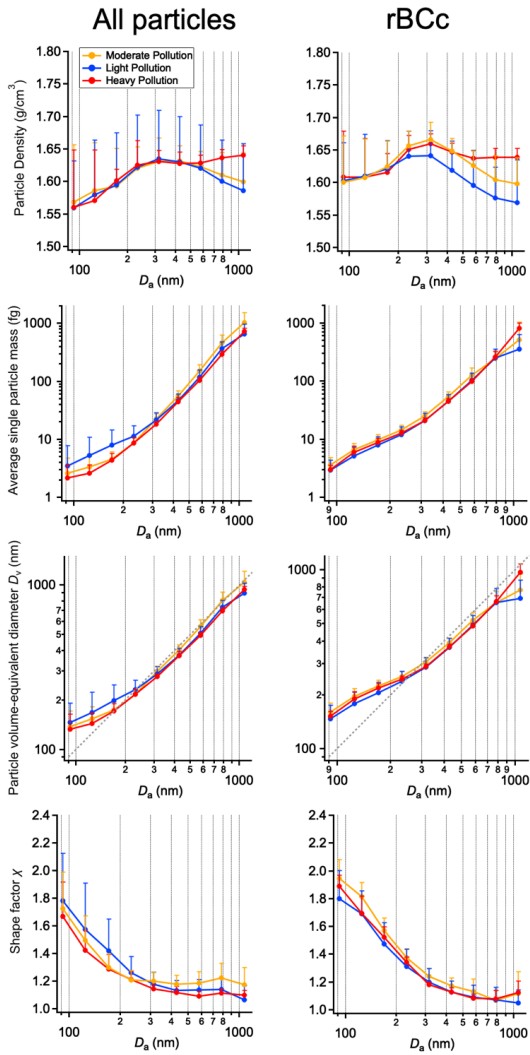


Figure 6 The particle density, average single particle mass, shape factor and volume-
equivalent diameter for all particles (All Particles, left) and refractory Black Carbon
containing particles (rBCc, right) under different pollution level.





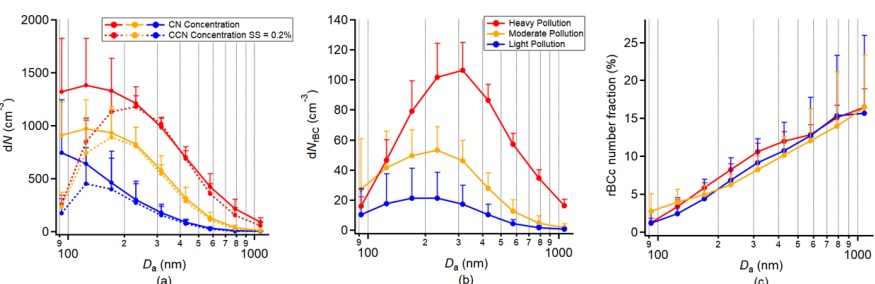


Figure 7 Size-resolved (a) CN and CCN at SS = 0.2% number concentrations; (b) rBC
number concentration; (c) rBCc number fraction.

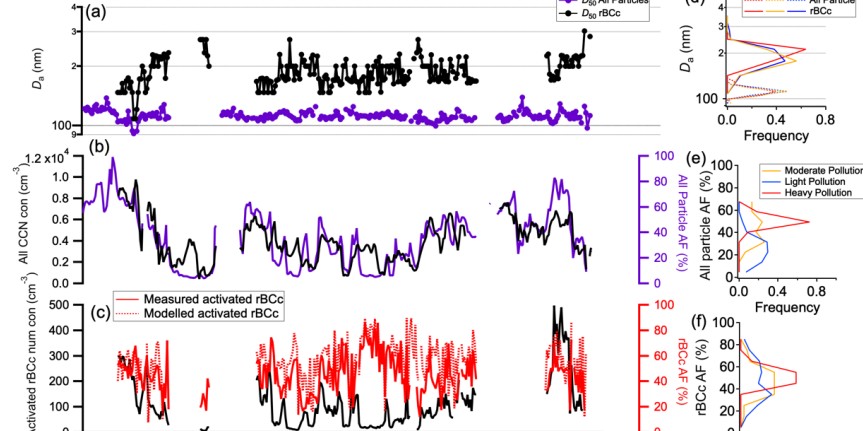


Figure 8 CCN activities of all particles and rBCc. (a) Time series of $D_{50}$ for all particles
and rBCc; (b) Time series of all CCN number concentrations and all particles activation
fractions; (c) Time series of measured activated rBCc number concentrations, and rBCc
activation fractions from two methods; (d) Frequency of $D_{50}$ for all particle and rBCc;
(e) Frequency of all particles activation fraction; (f) Frequency of measured rBCc
activation fraction.





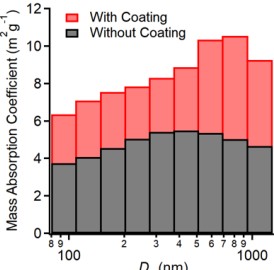


Figure 9 Mass Absorption Coefficient (MAC) at 880 nm wavelength for coated and

uncoated rBCc.
