# Peer review of "Aerodynamic size-resolved composition and cloud 1 condensation nuclei properties of aerosols in Beijing 2 suburban region 3 4 Chenjie Yu1,2, Dantong Liu1, Kang Hu1, Ping Tian3, Yangzhou Wu1, Delong Zhao3, 5 Huihui Wu2,"

_Atmospheric Chemistry and Physics, 2021_

## Author Response (AR1)

Firstly, we would like to thank both referees for their important comments, we have addressed all the comments below. The original comments from referees are in black, our replies are in blue and the changes in original manuscript are in red.

**Anonymous Referee #1**

This paper presents online, size-resolved measurements of black-carbon-containing particles and associated properties collected in a suburb of Beijing. The strength of this work is that it simultaneously probes size-resolved composition and other properties, which makes this work potentially useful for modeling studies in terms of providing ground truth of "what is out there". The work fits well within the scope of ACP. I have two major comments and a number of minor comments that should be addressed before the paper can be accepted for publication.

We thank the referee for the positive comments.

Major comment:
1. Section 2.2: An important assumption for the calculation of density and D_v,all is that mineral dust is not present in the samples, since dust cannot be detected by the instrumentation used for this study. This should be mentioned here, and a discussion is needed to what extent this assumption can be justified and what error is introduced by this assumption.

   We thank the referee for the comment. We consider that ignoring mineral dust particles will only have very limited impact on our results:

   Firstly, it is known that miner dust particles are mainly external mixed in coarse mode with aerodynamic diameter larger than 1 μm (Seinfeld and Pandis, 2016), while our study focuses on the $PM_1$ particles in fine particle mode (accumulation mode). Previous study reported that mineral dust particles only accounted around 10% of total $PM_1$ mass concentration in Beijing (Zhang et al., 2018), and most of them are with aerodynamic diameter larger than 500 nm (Li et al., 2014). Secondly, while this area is sometimes heavily influenced by dust transport from northern China and Mongolia, this did not occur during our measurement periods. We have included the following discussion:

   Mineral dust particles are not measured during the experiment because of the instrument upper detection limit. Mineral dust particles are mainly externally mixed with other aerosol compositions and mostly in coarse mode (with $D_a > 1$ μm) (Seinfeld and Pandis, 2016), and there was no dust event during the experimental periods. Previous measurements showed that mineral dust particle only accounted ~10% of total $PM_1$ mass concentration in Beijing (Zhang et al., 2018), and most of them were at $D_a > 500$ nm (Li et al., 2014). Therefore, our results may slightly underestimate the $\rho_{all}$ and total particle mass ($M_{all}$) in bins with $D_a > 500$ nm if dust is present, although we consider this effect to be minor.

2. The large, heavily coated particles reported here are interesting. I would assume that it takes a long time to grow such thick coatings on such large cores by condensation of low-volatile vapors. The authors mention coagulation as another possibility of how such these particles could be formed. However, coagulation is usually also a slow process in the atmosphere (depending on number concentrations). Could such thick coatings be formed by in-cloud aqueous phase chemistry and resuspension of the coated aerosol after the cloud evaporates?

Are the coatings of these particles really secondary in nature or are they possible at least partially primary? It would strengthen the paper if the authors provide could some back-of-the-envelope estimates of how these particles could be formed.

We thank the referee for the interesting comments regarding to the potential aging process.

We believe that this increasing trend of BC number fraction is likely attributed to coagulation after emission. As mentioned by Referee #1 that the coagulation rate is depending on the particle number concentrations. According to the measurements done by Zhang et al. (2020), they observed an increase in BC number concentrations with increasing BC core size for fresh firewood burning emission in Beijing surrounding rural regions. Due to the high concentration of BC particles and co-emitted NR particles, the coagulation of BC is rapid, and thickly coated rBCc with relatively large rBC core formed shortly after emission. Liu et al. (2019) also reported that the aging degree of rBCc is higher during winter heating period in North China than in summer.

We agree with the Referee #1's comment that in-cloud and resuspension process may be another possibility to form these larger rBCc. This process may lead larger rBC core size from the collision of cloud droplets (Ding et al., 2019), and thicker coating material due to in-cloud aqueous phase reactions. However, while this is an important hypothesis, we cannot test this here because it requires lots of assumptions regarding the air mass history and aerosol properties at the emission point.

We have revised the discussion here:

This tends to represent a generic phenomenon for a suburban environment with continuous influence of anthropogenic emissions, and the primary emissions had been aged in a time scale of hours. Fine rBC coagulated with pre-existing larger particles during the aging process (Riemer et al., 2009). The coagulation process dominated the formation of thickly-coated rBC particles (Reddington et al., 2013), and the coagulation rate of smaller rBC may be fast due to the higher number concentration of fine mode particles (Matsui et al., 2018). The coagulation occurs more rapidly near source due to the higher number concentration (Jacobson, 2005). Zhang et al. (2020) reported an increase in BC number concentrations with increasing rBC core size for fresh residential firewood burning emission in Beijing surrounding regions. Due to the high concentration of both rBC and co-emitted NR particles, the coagulation of rBC is rapid and thickly coated rBCc with relatively large rBC core formed shortly after emission. Liu et al. (2019) also reported higher aging degree of rBCc during the heating season in North China. However, the very fresh fossil fuel emissions such as from diesel engine emissions, which mostly contains thinly-coated small rBC (Han et al., 2019), may not show the same rBC number fraction distribution. Previous studies also reported a relatively fast aging process for BC also in the order of hours (Peng et al., 2017), if under a polluted environment rich of precursors, the aging could be even faster (Peng et al., 2016). The cause of this increased rBC number fraction at larger particle size is therefore the non-rBC compounds associated with it. The results presented here indicated that the higher contribution from regional pollution to the rBC number at larger aerodynamic size may apply, albeit the various features of primary sources in winter (Wang et al., 2019; Liu et al., 2019).

It is also possible that these larger rBCc may have experienced in-cloud processing. If the air parcel has passed through a cloud, the large and thickly coated rBCc are expected to

have be scavenged through activation, and the size of rBC core may increase effectively within the cloud because of the cloud droplet collision (Ding et al., 2019). Through in-cloud aqueous reactions, sulphate or organic matter may be added to the rBCc (Zhang et al., 2017). When the cloud dries out through the cloud evaporation or the air parcel descending in a downdraft, the core size and coatings may be enlarged for these released rBCc. Although the results presented here cannot alone test this hypothesis.

Minor comments:

1. L. 36: "By applying the mixing state of refractory black carbon containing particles …". The word "applying" sounds strange here. Do you mean "considering"?

   Accept. We have revised the revised the related sentences:

   By considering the mixing state of refractory black carbon containing particles (rBCc) and composition-dependent densities

2. L. 73-81: A relevant reference to motivate the work presented in this paper may be Ching J, Kajino M. Aerosol mixing state matters for particles deposition in human respiratory system. Scientific reports. 2018 Jun 11;8(1):1-1.

   Thanks for the suggestion. We have included the suggested publication in the background information:

   Fine particulate matter can also cause damage to human health via the respiratory system (Xing et al., 2016;Xu et al., 2016). The aerodynamic size of aerosols crucially determines the area of deposition (Sturm, 2010;Vu et al., 2018;Sturm, 2017), e.g. particles with an aerodynamic diameter ($D_a$) below 2.5 µm can reach the alveoli in the lungs and possibly pass into the blood (Lipworth et al., 2014). Through particle-resolving model simulation, Ching and Kajino (2018) found that the deposition efficiency in human alveoli also depended on the mixing state. The toxicity of aerosols is composition dependent (Kwon et al., 2020) and influenced by the complex morphology (Sturm, 2010; Vu et al., 2018), therefore the aerodynamic size-resolved properties of aerosols are important when understanding their influences on human health.

3. L. 150: Since different kinds of diameters are used in this paper depending on measurement technique, please specify what kind of diameters are Dp and Dc. It would be good if the results found for Beijing could be contrasted to other environments, for example see Motos G, Schmale J, Corbin JC, Zanatta M, Baltensperger U, Gysel-Beer M. Droplet activation behaviour of atmospheric black carbon particles in fog as a function of their size and mixing state. Atmospheric Chemistry and Physics. 2019 Feb 20;19(4):2183-207 and Motos G, Schmale J, Corbin JC, Modini R, Karlen N, Bertò M, Baltensperger U, Gysel-Beer M. Cloud droplet activation properties and scavenged fraction of black carbon in liquid-phase clouds at the high-alpine research station Jungfraujoch (3580 m asl). Atmospheric Chemistry and Physics. 2019 Mar 25;19(6):3833-55.

   Thanks for the comments. Combining with the Minor comments #7 we have revised the description of SP2:

   The minimum mass-equivalent diameter of rBC core that can be detected by SP2 is 70 nm using a rBC material density of 1.8 g/cm$^3$ . The volume properties of non-refractory

material within rBCc (hence referred as 'coating thickness') was derived by SP2 leading-edge-only (LEO) method and is described as the ratio between the optical volume-equivalent diameter of total rBCc and the mass-equivalent diameter of rBC core ($D_p/D_c$) (Liu et al., 2019). Previous morphology-independent measurement study validated that the uncertainty of the SP2 LEO fitting method in determining the rBCc coating thickness is within 20% (Yu et al., 2020).

We thank the referee for the suggestions of studies in other environments. We have included the BC activation results in Switzerland:

Our results confirm that while rBCc can be CCN active, the size of rBCc is crucial to the rBCc CCN ability in polluted suburban environment. This agrees with the previous study done by Motos et al. (2019b) who also found that the size of rBCc is important for rBCc activation at certain SS. The $AF_{rBCc}$ result presented in our study is generally consistent with previous field measurements in anthropogenic polluted environments in China: Wu et al. (2019) and Hu et al. (2021a) reported 59% and 60% of total rBCc could be activated at SS = 0.2% respectively. Studies performed in other environments also found that coated BC can be CCN active: Motos et al. (2019a) and (2019b) reported that ~6%-12% and ~40%-70% of total BC mass fraction can be activated with SS ≈ 0.05% in Zurich and SS ≈ 0.2% at Jungfraujoch respectively.

4. Section 2.2: The notation in this section is unclear. Equation (1) refers to a per-particle quantity, but this is not how the authors use it here, since Line 179 mentions "for all particles", which I assume means "averaged over all particles". However, Figure 6 presents size-resolved graphs of shape parameter, which implies that equation (1) is calculated as an average for each size bin. Please clarify and improve the notation so that this becomes clear.

Thanks for the comments. We have revised this section to make it clear. We have revised the Section title as:

2.2 Calculation of size-resolved aerosol morphology parameters

We have revised the description of Equation (1):

Based on the size-resolved measurement here, $\chi$ at each AAC selected size bin can be calculated by:

$$\chi = \frac{\rho_p D_v{}^2 C_c(D_v)}{D_a{}^2 C_c(D_a)} \tag{1}$$

And we have revised the description in Line 179:

This calculation is performed for all particles (including rBCc) and rBCc at each size bin, using their respective parameters ($\rho_p$ and $D_v$).

5. Line 276: "To test whether the mixing state varies according to ambient pollution concentrations…" Do you mean mixing state as in "internally/externally mixed" (i.e., within a given size range)? This sounds like an interesting idea, but I don't think it is

answered in the results section since this is looking at the dependence of various quantities on particle size – I would not call this mixing state.

Thanks for the comment. We have revised the description here:

To test whether the aerosol physiochemical properties vary according to ambient pollution concentrations

6. Line 292: "no significant difference": suggest rephrasing this since it is not meant in the sense of "statistically significant" Section 3.2: It would help the reader if you could add to this section references to the individual figures that you are referring to.

Thanks for the comments. We have revised the description here to avoid further misunderstood:

Though the heavy pollution period has the highest aerosol mass concentrations among three cases, the mass concentrations for total $PM_1$ and non-refractory compositions were close at $D_a < 200$ nm under different pollution levels.

Following the referee's suggestion, we have added the figure number for each part of the description:

Though the heavy pollution period has the highest aerosol mass concentrations among three cases, the mass concentrations for total $PM_1$ and non-refractory compositions were close at $D_a < 200$ nm under different pollution levels (Fig. 5(a)). Notable contributions to the total $PM_1$ from non-refractory material was observed for $D_a > 300$ nm especially for the heavy pollution condition in Fig. 5(i-k). Unlike the more polluted conditions, the non-refractory aerosol mass concentrations during light pollution periods shows limited size-dependent variation. Fig. 5(b) shows that the size distribution of rBC mass concentration reached the peak at 400 nm under heavy pollution, while the peak for the moderate and light pollution was at a smaller $D_a$ which was between 300 and 400 nm. Fig. 5(c-e) show that the peak diameter of NR-$PM_1$ observed in Pinggu was at around 700 nm and is higher than the peak diameter of NR-$PM_1$ reported at the urban site of Beijing which is between 400 nm and 500 nm in winter.

7. Definition of rBC-containing particles: What is the minimum core size that can be detected?

Accept. We have added the detection range of SP2 in the Experimental methods section. Please refer to our response to Minor comment #3 for the revised context.

8. Line 374: Is the more spherical-like morphology for larger particles because the coating material forms a spherical coating around the non-spherical core, or does the core itself become more spherical because it collapses? This may seem a minor point, but it is important when justifying the use of Mie calculations for BC-containing particles, which assumes a spherical core.

Thanks for the comments. We think the rBCc become more spherical-like due to the coating material forms a spherical coating. We have revised the discussions here:

This result indicates that smaller particles have more irregular shapes, while particles with larger aerodynamic size are more spherical in ambient atmosphere. Previous experiments have shown that the irregular rBCc from fresh emissions can transform to be more spherical-like by acquiring more secondary substances (Ahern et al., 2016). Hu et al. (2021b) illustrates that the acquisition of coating material is more important for the overall rBCc shape, while the shape of rBC core is not sufficient to describe the change of overall rBCc shape.

9.  Line 470: Why were optical properties evaluated for a wavelength of 880 nm? Unless this was done to compare to measurements (which is not the case here), it is more common to do this for ~500 nm (closer to the peak of the spectrum from sunlight).

    Thanks for the comment here. We have included the Mass Absorption Coefficient (MAC) at 550 nm derived through the core-shell Mie theory as Fig 9(a) and revised the related discussions:
    As indicated in Fig. 9, the mass absorption coefficient (MAC) of rBCc at 550 nm and 880 nm wavelength is calculated through the core-shell Mie model described in the supplementary. The $MAC_{550}$ and $MAC_{880}$ was about 2-fold enhanced for rBCc with $D_a >$ 500 nm. These larger rBCc with high absorption efficiency importantly contributed to the total absorption.

[Figure]

Figure 9 Mass Absorption Coefficient (MAC) at (a) 550nm (b) and 880 nm wavelength for coated and uncoated rBCc.

10. The use of English language is appropriate for the most part although some paragraphs/sentences would benefit from being proof-read by a native English speaker.

    Accept. We have done another round of proof-reading to improve the presentation quality of our manuscript.

**Anonymous Referee #2**

This manuscript by Yu et al. introduced the aerodynamic size-resolved chemical composition and CCN activity of aerosols in the Beijing suburban region. The study combined an aerosol aerodynamic classifier (AAC) with a set of aerosol physical and chemical measurements and focused on the properties of refractory black carbon-containing particles (rBCc). The study found that rBCc are relatively spherical at sizes above 300 nm, and the number fraction of rBC increases as a function of particle size. Due to the coating properties and their larger sizes, a relatively large fraction of the rBCc could also be activated to contribute to cloud formation.

The manuscript is well organized. I recommend the publication of the manuscript after the following minor revisions.

We thank the referee for the positive comments.

General comments:

1. I found the higher number fraction and larger MMD of rBC at larger sizes very interesting. The authors attributed this phenomenon to particle coagulation. Is this coagulation happening among the rBC particles or between rBC and other larger particles? Considering that fresh soot particles directly generated from engines are relatively small (~ 100 nm), how fast is this coagulation process and how does the involvement of other chemical species affect the evolution of rBC?

   We thank the referee's interest in our results. Reflected in Referee #1's major comment and our response, the coagulation rate is depending on the particle number concentrations (Jacobson, 2005). Though the fresh soot particles may be relatively small, they may coagulate with larger particles, thus increasing the ratio of non-BC to BC mass. In addition to the engine emitted soot mentioned by the Referee #2, we think residential biofuel burning is another important source during the winter period in North China. The fast coagulation process presented in Zhang et al. (2020) shown that the residential biofuel burning emitted rBCc can be thickly coated and with relatively large rBC core size shortly after emission. We have expanded our discussions here following both referee's useful suggestions. Please refer to our response to Referee #1's major comment 2 for the revised context.

   But as mentioned in our response to Referee #1, it is difficult to explain the aging process completely here since it needs lots of assumptions regarding to the air mass history and initial aerosol properties.

2. The coating thickness Dp/Dc showed a minimum value at the size of around 300 nm. The authors imply that traffic emissions may play a role in this change of coating properties. Could the authors elaborate more on the detailed mechanisms?

   We thank the referee's comment. We have revised the discussion here:

   This decreasing trend of rBC coating thickness may be caused by the local traffic emissions. Studies have shown that fresh traffic diesel engine emitted rBCc importantly contribute to the condensation mode particles with diameters of 100-300 nm (Seinfeld and Pandis, 2016; Gong et al., 2016). Joshi et al. (2021) demonstrated that traffic emissions dominated the rBC fluxes in Beijing, and previous studies also showed a similar decreasing trend of rBC coatings for engine emissions within this particle size range (Han et al., 2019; Zhang et al., 2020). According to the diffusion-controlled particle growth law, smaller particles diffuse more quickly and hence grow more effectively than the larger particles (Seinfeld and Pandis, 2016). Therefore, smaller rBC acquire more coatings within particle diameters of 100-300 nm.

Detailed comments:

1. Page 4 Line 103: There are a few more studies on size-resolved CCN activity and aerosol physiochemical properties in Beijing, such as the following ones. Probably the authors want to stress that this study focused on rBC and used an AAC to size-classify the aerosols.

We thank the referee for the useful suggestions. We have revised the statement here and included the suggested publications:

Previous studies (Levin et al., 2014; Broekhuizen et al., 2006; Gunthe et al., 2011; Fan et al., 2020; Wu et al., 2017) have provided both the measured size-resolved CCN ability and aerosol physiochemical properties. However, the CCN ability of rBCc based on atmospheric data remains poorly constrained.

2. Page 4 Line 113: One of the disadvantages for the AMS measuring the size-resolved composition is that the AMS cannot measure CCN related sizes (50 to 100 nm). But it appears that the AAC was used in size range of 90 to 1100 nm, which is not significantly better than the AMS measurement range.

We appreciate the referee's comment, but we think there may be a misunderstanding here. We are trying to highlight that the AAC combination used in our study is able to perform size-resolved measurements for both the CCN ability (from CCNc) and the aerosol compositions (from AMS+SP2) at the same time. Though the AMS p-ToF mode alone can also deliver the vacuum aerodynamic size-resolved aerosol compositions, it cannot measure the size-resolved CCN concentrations at the same time and may not be suitable for the composition based CCN ability derivation due to its poor accuracy at certain size range. We have revised the related description to make it clear:

The aerosol mass spectrometer (AMS) is also capable of delivering size-resolved aerosol compositions, however the poor accuracy of AMS in the size range important for CCN (typically 50 -100 nm) has hampered quantitative work for the application of CCN concentration derivation based on $\kappa$-Köhler theory (Petters and Kreidenweis, 2007).

3. Page 7 Eq. (2): How was rou_NR calculated? It may be better to include a table of nomenclature to introduce each of the parameters and how they are measured (by which instrument) or calculated.

Thanks for the comment. Firstly, the coating material of rBCc is assumed to constitute of the same volume fractions of ambient non-refractory compositions following the previous studies(Liu et al., 2015; Hu et al., 2021a; Motos et al., 2019b). Then $\rho_{NR}$ is derived following the equation:

$$\rho_{NR} = \frac{M_{NR}}{V_{NR}}$$

where $M_{NR}$ is from the total AMS measurement results. We applied an ion pairing scheme introduced by Gysel et al. (2007) for the AMS measurement results which is also included in our supplementary:

$$n_{NH_4NO_3} = n_{NO_3^-}$$

$$n_{H_2SO_4} = \max\left(0, n_{SO_4^{2-}} - n_{NH_4^+} + n_{NO_3^-}\right)$$

$$n_{\text{NH}_4\text{HSO}_4} = \min\left(2n_{\text{SO}_4^{2-}} - n_{\text{NH}_4^+} + n_{\text{NO}_3^-}, n_{\text{NH}_4^+} - n_{\text{NO}_3^-}\right)$$

$$n_{(\text{NH}_4)_2\text{SO}_4} = \max\left(n_{\text{NH}_4^+} - n_{\text{NO}_3^-} - n_{\text{SO}_4^{2-}}, 0\right)$$

$$n_{\text{HNO}_3} = 0$$

where $n$ represents the number of moles for each component. Then $V_{\text{NR}}$ is derived through the mass concentration and the material density (presented in Table S1) of each measured NR composition:

$$V_{\text{NR}} = \frac{M_{(\text{NH}_4)_2\text{SO}_4}}{\rho_{(\text{NH}_4)_2\text{SO}_4}} + \frac{M_{\text{NH}_4\text{NO}_3}}{\rho_{\text{NH}_4\text{NO}_3}} + \frac{M_{\text{NH}_4\text{HSO}_4}}{\rho_{\text{NH}_4\text{HSO}_4}} + \frac{M_{\text{H}_2\text{SO}_4}}{\rho_{\text{H}_2\text{SO}_4}} + \frac{M_{\text{Org}}}{\rho_{\text{Org}}}$$

We have included Table as Appendix A to better describe this:

**Appendix A Parameters used for the calculation of size-resolved shape factors**

| Parameter | Description | Calculation/Measurement | References |
|---|---|---|---|
| $M_{\text{NR}}$ | Mass concentration of total non-refractory compositions | Sum of AMS results | |
| $M_{\text{rBC}}$ | Mass concentration of rBC | SP2 measurement | |
| $M_{\text{all}}$ | Mass concentration of all particles | Sum of AMS and SP2 results | |
| $N_{\text{total}}$ | Number concentration of all particles | CPC measurement | |
| $V_{\text{all}}$ | Volume of all particles | $V_{\text{all}} = \dfrac{M_{(\text{NH}_4)_2\text{SO}_4}}{\rho_{(\text{NH}_4)_2\text{SO}_4}} + \dfrac{M_{\text{NH}_4\text{NO}_3}}{\rho_{\text{NH}_4\text{NO}_3}} + \dfrac{M_{\text{NH}_4\text{HSO}_4}}{\rho_{\text{NH}_4\text{HSO}_4}} + \dfrac{M_{\text{H}_2\text{SO}_4}}{\rho_{\text{H}_2\text{SO}_4}} + \dfrac{M_{\text{Org}}}{\rho_{\text{Org}}} + \dfrac{M_{\text{rBC}}}{\rho_{\text{rBC}}}$ | (Gysel et al., 2007; Hu et al., 2021a) |
| $V_{\text{NR}}$ | Volume of non-refractory compositions | $V_{\text{NR}} = \dfrac{M_{(\text{NH}_4)_2\text{SO}_4}}{\rho_{(\text{NH}_4)_2\text{SO}_4}} + \dfrac{M_{\text{NH}_4\text{NO}_3}}{\rho_{\text{NH}_4\text{NO}_3}} + \dfrac{M_{\text{NH}_4\text{HSO}_4}}{\rho_{\text{NH}_4\text{HSO}_4}} + \dfrac{M_{\text{H}_2\text{SO}_4}}{\rho_{\text{H}_2\text{SO}_4}} + \dfrac{M_{\text{Org}}}{\rho_{\text{Org}}}$ | (Gysel et al., 2007; Hu et al., 2021a) |
| $D_{\text{c}}$ | Mass equivalent diameter of rBC core | SP2 measurement | |
| $D_{\text{p,rBCc}}$ | Volume equivalent diameter of rBC-containing particle | SP2 LEO fitting method, assumed to be equal to the optical diameter | (Liu et al., 2019) |

| $\rho_{NR}$ | Non-refractory aerosol composition density | $\rho_{NR} = \dfrac{M_{NR}}{V_{NR}}$ | (Hu et al., 2021a) |
| $\rho_{all}$ | All particles density | $\rho_{all} = \dfrac{M_{all}}{V_{all}}$ | |

4. Page 7 Eq. (5): How was epsilon_coating,i measured or calculated? Also, in Fig. 5h, the kappa values for rBCc are sometimes higher than those of all particles. Is this reasonable?

Thanks for the comment. The volume of the coating material is derived through the SP2 LEO fitting method, and the coating material of rBCc is assumed to constitute of the same volume fractions of ambient non-refractory compositions measured by AMS following the previous studies (Motos et al., 2019b; Hu et al., 2021a).

In Fig. 5h, the result is reasonable since the calculation of kappa values for all particles ($\kappa_{all}$) ignore the NR-coating within the rBCc as described in function (4). When it comes to larger particle size ($D_a$ > 600nm), the hydrophobic rBC core only accounted limited volume fraction within the total rBCc, and the rBCc become more hydrophilic. However, the calculation of $\kappa_{all}$ ignores the thick coating material of rBCc. Therefore, the $\kappa_{all}$ may be slightly underestimated. As the values of $\kappa_{all}$ only act as a reference in our study, this underestimation will not influence our following results and conclusions. We have included this uncertainty information in Section 2.3:

Due to the coating material of rBCc is not included in the calculation process of $\kappa_{all}$ here, $\kappa_{all}$ may be slightly underestimated when rBCc is thickly coated at larger particle size.

5. Page 9 Line 252: "idea" change to "ideal"

Accept. We have fixed the typo here:

$R$ is the ideal gas constant.

6. Page 9 Line 254: Was the scanning of the SS done in this work? In the methods section, the SS was fixed at 0.2%.

We appreciate the referee's suggestion. But the SS of the CCNc was set to be fixed at SS=0.2% in this study.

7. Page 10 Line 296: "… non-refractory aerosol mass concentrations during light pollution periods show limited size-dependent variation." There are indeed significant variations as a function of particle size, although the absolute concentrations are relatively low.

Thanks for the comment. We have revised the description here:

Unlike the more polluted conditions, the non-refractory aerosol mass concentrations during light pollution periods shows size-dependent variation in a much lower absolute value range.

8. Page 10 Line 306: Why was the Org highly oxidized in Beijing suburban? It may be better to elaborate this observation in more details.

Thanks for the suggestion. We have revised the discussion here:

The Org peak diameter in Pinggu is at around 700 nm which is close to the peak diameter of secondary inorganic compositions. The average oxygen to carbon ratio (O/C ratio) for the total Org aerosol in Pinggu is 0.5 and is higher than the O/C ratio in Beijing urban region in winter which is 0.32 reported by Hu et al. (2016). These suggest the higher oxidization of Org in Beijing suburban than the urban region.

9. Page 13 Line 400: "Fine rBC condensed …" Since rBC are in the particle phase, not in the vapor phase, they cannot "condense" onto pre-existing particles.

Accept. We have fixed the related sentence:

Fine rBC coagulated with pre-existing larger particles during the aging process

10. Page 15 Line 456: remove "and".

Accept.

Our results confirm that while rBCc can be CCN active, the size of rBCc is crucial to the rBCc CCN ability in polluted suburban environment.

11. Fig. 5b, 5c, 5d, 5e and 7a, 7b: Should the labels on the y-axes "dM" and "dN" be "dM/dlogDp" and "dN/dlogDp"? Also, why are the error bars shown in the positive direction only?

Thanks for the comments. We have revised the Fig. 5 and Fig. 7 following the comments. We have also fixed some minor errors in Fig. 7. We appreciate the referee's suggestions for the error bars, we only present the error bars in the positive direction to avoid overlap and keep the graphs neat and easier to read.

[Figure]

12. Fig. 8: Please show the legends in panels b and c (black curve is not introduced).

Accept. Fig.8 has been revised as following:

[revised manuscript text omitted]